# Quality of Life among Patients with Acute Coronary Syndromes Receiving Care from Public and Private Health Care Systems in Brazil

Ingrid Maria Novais Barros de Carvalho Costa [1,2], Danielle Góes da Silva [3],
Joselina Luzia Meneses Oliveira [1,4,5,6], José Rodrigo Santos Silva [7], Fabrício Anjos de Andrade [8],
Juliana de Góes Jorge [1], Larissa Marina Santana Mendonça de Oliveira [1], Rebeca Rocha de Almeida [1,*],
Victor Batista Oliveira [1], Larissa Santos Martins [9], Jamille Oliveira Costa [1], Márcia Ferreira Cândido de Souza [10],
Larissa Monteiro Costa Pereira [1], Luciana Vieira Sousa Alves [1], Silvia Maria Voci [3],
Marcos Antonio Almeida-Santos [5,11], Felipe J. Aidar [12,*], Leonardo Baumworcel [1,5]
and Antônio Carlos Sobral Sousa [1,4,5,6]

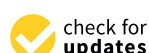



[1] Graduation Program in Health Sciences, Federal University of Sergipe, São Cristóvão 49100-000, Brazil; ingrid_novais@infonet.com.br (I.M.N.B.d.C.C.); joselinamenezes@gmail.com (J.L.M.O.); julianagoesfisio@yahoo.com.br (J.d.G.J.); nutrilarissamarina@gmail.com (L.M.S.M.d.O.); vbo.nutri@gmail.com (V.B.O.); jamillenutri@gmail.com (J.O.C.); larissaa_monteiroo@hotmail.com (L.M.C.P.); lucianaalvesnutri@gmail.com (L.V.S.A.); leonardo.baumworcel@caxiasdor.com.br (L.B.); acssousa@terra.com.br (A.C.S.S.)
[2] Federal Institute of Sergipe, São Cristóvão 49100-000, Brazil
[3] Department of Nutrition, Federal University of Sergipe, São Cristóvão 49100-000, Brazil; danygoes@academico.ufs.br (D.G.d.S.); smvoci@academico.ufs.br (S.M.V.)
[4] Department of Medicine, Federal University of Sergipe, São Cristóvão 49100-000, Brazil
[5] São Lucas Clinic and Hospital/Rede D'Or São Luiz, Aracaju 49015-380, Brazil; marcosalmeida2010@yahoo.com.br
[6] Division of Cardiology, University Hospital, Federal University of Sergipe, Aracaju 49060-025, Brazil
[7] Department of Statistics and Actuarial Sciences, Federal University of Sergipe, São Cristóvão 49100-000, Brazil; rodrigo.ufs@gmail.com
[8] Primavera Hospital, Aracaju 49026-010, Brazil; fabricioanjos@globo.com
[9] Graduate Program in Nutrition Sciences, Federal University of Sergipe, São Cristóvão 49100-000, Brazil; martins.lss@outlook.com
[10] Division of Nutrition, University Hospital, Federal University of Sergipe, Aracaju 49060-025, Brazil; nutrimarciacandido@gmail.com
[11] Graduate Program in Health and Environment, Tiradentes University, Aracaju 49032-490, Brazil
[12] Group of Studies and Research in Performance, Sport, Health and Paralympic Sports–GEPEPS, Federal University of Sergipe, São Cristóvão 49100-000, Brazil
[*] Correspondence: rebeca_nut@hotmail.com (R.R.d.A.); fjaidar@academico.ufs.br (F.J.A.)

**Abstract:** (1) Background: Quality of life (QOL) is used as a health indicator to assess the effectiveness and impact of therapies in certain groups of patients. This study aimed to analyze the QOL of patients with acute coronary syndrome (ACS) who received medical treatment by a public or private health care system. (2) Methods: This observational, prospective, longitudinal study was carried out in four referral hospitals providing cardiology services in Sergipe, Brazil. QoL was evaluated using the Medical Outcomes Study 36-Item Short-Form Health Survey. The volunteers were divided into two groups (public or private health care group) according to the type of health care provided. Multiple linear regression models were used to evaluate QoL at 180 days after ACS. (3) Results: A total of 581 patients were eligible, including 44.1% and 55.9% for public and private health care, respectively. At 180 days after ACS, the public health care group had lower QoL scores for all domains (functional capacity, physical aspects, pain, general health status, vitality, social condition, emotional profile, and health) ($p < 0.05$) than the private group. The highest QoL level was associated with male sex ($p < 0.05$) and adherence to physical activity ($p \leq 0.003$) for all assessed domains. (4) Conclusions: This shows that social factors and health status disparities influence QoL after ACS in Sergipe.

**Keywords:** physical activity; secondary prevention; quality of health care

## 1. Introduction

Acute coronary syndrome (ACS) is one of the most important causes of morbidity and mortality in Brazil and worldwide [1,2]. Despite the progress in the diagnosis and treatment of patients with ACS, which have contributed to a significant increase in the number of survivors after an acute event, it is still a challenge for health systems to provide effective, equitable secondary prevention measures [3–6] and addressing disparities in health care system for these patients.

Brazilian [1] and international [7,8] guidelines point to the importance of adequate secondary prevention guidance in patients with ACS. Prognosis and clinical evolution of patients after hospital discharge can be modified based on the therapy adopted and compliance to treatment, contributing to a reduction and control of risk factors (RF) and comorbidities, collaborating to an increase in survival [7–9] and improvement in the quality of life (QoL) of these patients [10].

QoL has become one of the most discussed topics in recent decades and is considered to be of great interdisciplinary interest nowadays [9,11,12], since the improvement in QoL has become an outcome of aftercare practices and public policies for health promotion and disease prevention [11,12]. Therefore, information about QoL has been used as an indicator to assess the effectiveness and impact of determined treatments on groups of patients [11–14].

In Brazil, the Brazilian Unified Health System (SUS in Portuguese), with universal coverage for 150 million Brazilians, coexists with the Supplementary Health Care, which is predominantly a private system, with 50 million beneficiaries. SUS was developed to meet the principles of universality, equality, and integrality [15,16]. However, there are reports of existing disparities between private and public health care with regard to the appropriate treatment of patients with ACS [4,17]. Moreover, evidence shows that distortions in the quality of health care may have a negative influence on treatment adherence, compromising the prognosis and QoL of patients [18]. However, information is scarce in the literature on the QOL of patients with ACS assisted in the SUS or private health care, and on the presence of disparity between health care systems.

## 2. Materials and Methods

### 2.1. Study Design and Locations

This observational, prospective, longitudinal study was carried out in four referral hospitals providing cardiology services in Aracaju City, Sergipe, Brazil. Among these hospitals, only one offers services through SUS and does not have an "open-door" service, which means that it requires the referral of patients from another health institution. The other three hospitals only offer Private Health Care Service (PHCS), either through health insurance or disbursement.

Our research followed the components of the Strengthening the Reporting of Observational Studies in Epidemiology (STROBE) [19] protocol for observational studies, as shown in Figure 1.

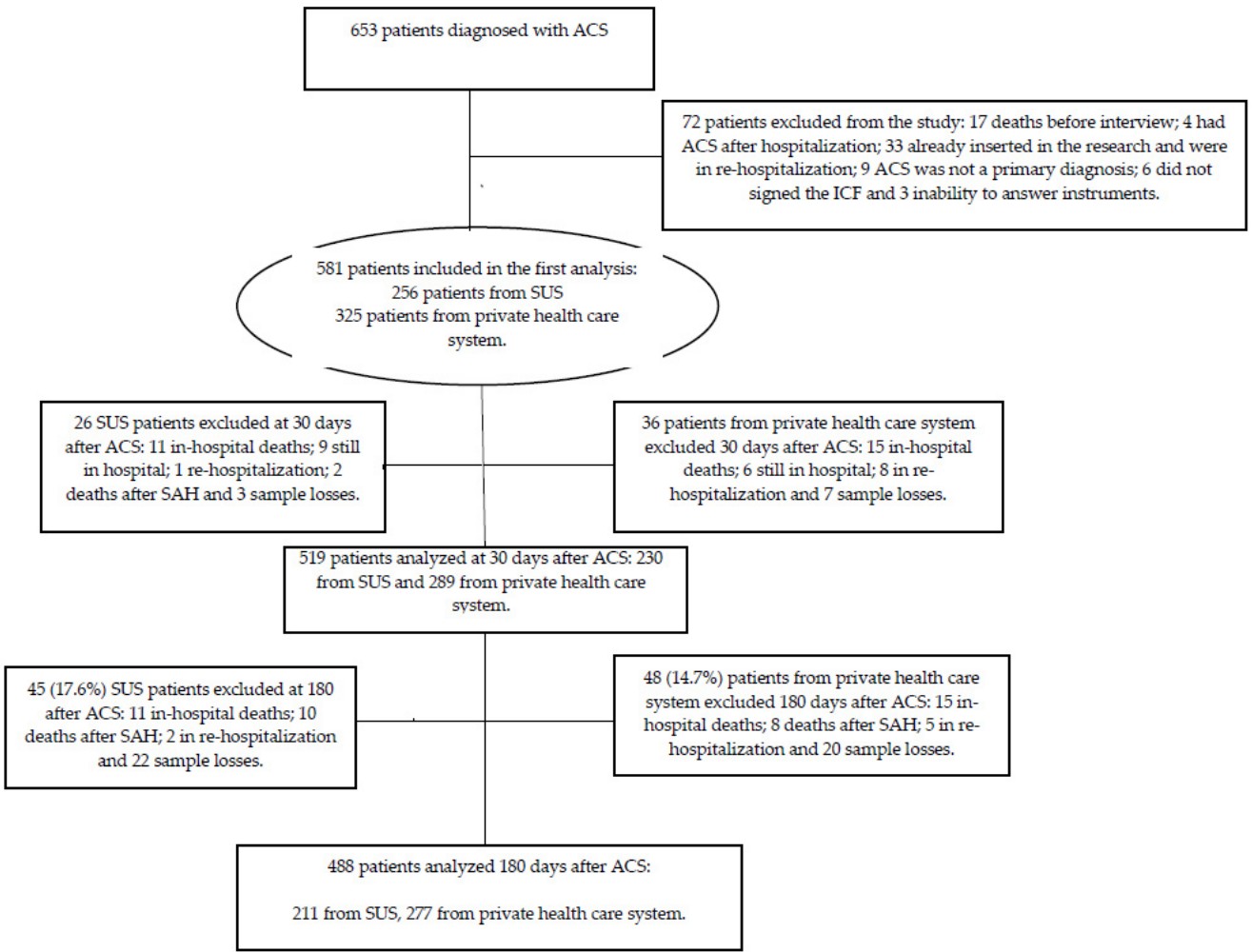

**Figure 1.** Study design.

*2.2. Study Sample*

We adopted the "all-comers"' sample type. This study enrolled 581 volunteers of both sexes, aged >18 years. They were consecutively diagnosed with ACS, which was characterized by unstable angina (UA), acute myocardial infarction (AMI) without ST-segment elevation (NSTEMI), or AMI with ST-segment elevation (STEMI). Patients who did not agree to participate in the study by signing the informed consent form and/or who were unable to answer the study protocols were excluded from the study. The inclusion and exclusion process is shown in Figure 1.

The diagnosis of ACS was based on the patients' clinical history, with the onset of consistent symptoms of acute ischemia during the previous 24 h, including or not a series of increases in myocardial necrosis markers. These data were confirmed by electrocardiography, Doppler echocardiography, or cine coronary angiography. In some cases, the diagnosis was confirmed using more than one of the previously cited examinations [20].

Our study was submitted to the Research Ethics Committee involving human beings at the Federal University of Sergipe (CEP/UFS). The committee approved our research (approval no. 302,544). All patients signed the informed consent form.

*2.3. Data Collection*

Data were collected from October 2013 to March 2016. The study consisted of three stages: (1) initial evaluation after the diagnosis of ACS (hospitalization); (2) follow-up assessment 30 days after ACS; (3) final evaluation at 180 days after ACS. To this end, we used the Case Report Form, which is composed of variables that provide information about

patients' sociodemographic and clinical conditions, levels of physical activity, quality of dietary intake, and QoL. To fill this form, data were obtained through interviews with the patient or one family member when the patients could not respond to the questionnaire by themselves. Their medical records were also analyzed.

The protocols of the medical teams of the hospitals followed national and international guidelines for patients with ACS [1,7–9]. At hospital discharge, individuals received general orientation regarding dietary intake, smoking cessation, physical activity, and adherence to drug treatment to prevent disease recurrence. The present study sought to verify the QoL of individuals, with the perspective that the greater the adherence to secondary prevention, the higher the QoL scores would be.

It is important to emphasize that at no time did the team of researchers of the study that originated this article perform interventions on the patients included in the research.

At admission and 180 days after ACS, the International Physical Activity Questionnaire (short version) [21,22] was used to assess adherence to physical activity recommendations. In addition, the Food Frequency Questionnaire [23] was used to collect information on dietary consumption, and the Alternative Healthy Eating Index (2010) [24] adapted from the Food Guide for the Brazilian Population [25] was used to assess their diet quality: the higher the values, the better the state of health. At 180 days after ACS, patients were surveyed about smoking cessation, and information on new cardiovascular events.

In the context of secondary prevention, some classes of medications are labeled as A according to the Specialized Guidelines [1], such as (a) antithrombotics: acetylsalicylic acid (ASA) and/or a P2Y12 inhibitor (Prazygrel, Ticagrelor or Clopidogrel); (b) β-blockers; (c) statins; (d) angiotensin-converting enzyme inhibitors (ACEI)/AT1 Receptor Blockers (ARB) and aldosterone receptor antagonist (spironolactone) in case of heart failure and/or left ventricular dysfunction. We collected the data related to the prescriptions of the medicines mentioned above from the medical records and compared them to the prescriptions, with the patients present at the moment of hospital discharge. Patients were considered adherent at 30 and 180 days post ACS when they reported using all prescribed medications.

About the socioeconomic level of the sample, according to the Brazilian Economic Classification Criterion of the Brazilian Association of Research Companies (ABEP) [26]. For purposes of analysis, the eight economic levels, or levels, or economic classes, established by ABEP, were regrouped and named as follows: A1, A2, and B1 in High Economic Level (A); B2, C1, and C2 in Medium Economic Level (M), and D and E in Low Economic Level (B).

To assess QoL, we applied the Medical Outcomes Study 36-Item Short-Form Health Survey (SF-36) questionnaire [27], since we used it in research with a specific focus on cardiology [28]. SF-36 consists of a self-administered instrument, which can also be part of an interview, whether face-to-face or by telephone [29], and SF-36 is composed of 36 questions that address eight domains in two major components: physical, which involves functional capacity, physical appearance, pain, and general health, and mental, which covers vitality, social aspects, emotional state, and mental health. We measured these domains in a score ranging from 0 to 100. The higher the score, the better the QoL. SF-36 also includes an item that assesses the individuals' perception of their own health compared to a year ago [28,29]. As regards the differences in patients in terms of their educational level, we decided to interview them to standardize our investigation. Therefore, a face-to-face interview was carried out at admission and by telephone at 30 and 180 days after the acute event.

### 2.4. Data Analysis

For data analysis, the patients were divided into two groups (SUS = public and PHCS = private health care groups) according to the type of health care received when they presented ACS. The distribution type of numerical variables was determined using the Kolmogorov–Smirnov test. Data with normal distribution were presented as means and standard deviations and categorical variables as absolute and relative frequencies (%).

The Mann–Whitney and Wilcoxon tests were applied to compare quantitative variables between groups for evaluation at different times and Friedman test for multiple comparisons. The association between groups and categorical variables was also verified using Pearson's Chi-square test or Fisher's exact test when appropriate.

To assess the internal consistency of the SF-36, Cronbach's alpha was calculated, which presented an average of 0.91, representing excellent reliability of the instrument. In addition, we developed a multiple linear regression model with the scores of the SF-36 domains at 180 days after ACS as dependent variables. The following independent variables were adopted: age, sex, educational level, type of health care, presence of comorbidities (systemic arterial hypertension (SAH), diabetes mellitus, dyslipidemia (DLP), overweight, and abdominal obesity), occurrence of a new cardiovascular event within 180 days after ACS, adherence to physical activity, adherence to pharmacotherapy, diet quality index, smoking cessation after 180 days of ACS, and hospitalization time (assessed by percentage variation). A 95% confidence interval was adopted for independent variables associated with the scores of the SF-36 domains. Statistical analyses were carried out using the R Core Team 2016 Program version 3.3.2, with the significance level used being 5%.

## 3. Results

A total of 581 patients were considered potentially eligible for the study: 256 (44.1%) received medical care from public health care and 325 (55.9%) from the private one. At 30 and 180 days after ACS, we interviewed 519 and 488 patients, respectively.

In general, the patient's baseline and adherence characteristics in the public health care group (SUS) differed from those in the private health care system. Patients in the public health care group were predominantly younger men, with lower socioeconomic status, higher prevalence of STEMI, alcoholism, smoking, and less adherence to secondary prevention treatment after ACS. Patients treated by the private health care system had more comorbidities, but with a shorter hospital stay. No distinction was found between the groups regarding the occurrence of cardiovascular outcomes at 180 days of ACS (Table 1).

**Table 1.** Baseline characteristics, cardiovascular outcomes, and adherence to secondary prevention (adherence to physical activity, medication, smoking cessation, and diet quality) in patients with ACS, according to the Type of Healthcare, Aracaju, Brazil.

| Categorical Variables | Valid Patients | Type of Healthcare | | *p* |
|---|---|---|---|---|
| | | SUS (%) | PHCS (%) | |
| Age Group (years) | | | | |
| from 18 to 49 | | 51 (19.9) | 24 (7.4) | |
| from 50 to59 | | 68 (26.6) | 76 (23.4) | |
| from 60 to 69 | 581 | 88 (34.4) | 109 (33.5) | <0.001 |
| from 70 to79 | | 38 (14.8) | 70 (21.5) | |
| ≥80 | | 11 (4.3) | 46 (14.2) | |
| Sex | 581 | | | |
| Male | | 181 (70.7) | 189 (58.2) | 0.002 |
| Schooling (years) | | | | |
| No schooling or <1 year | | 32 (12.5) | 12 (3.7) | |
| from 1 to 3 | 581 | 64 (25.0) | 20 (6.1) | <0.001 |
| from 4 to 8 | | 99 (38.7) | 79 (24.3) | |
| 9 years or more | | 61 (23.8) | 214 (65.9) | |

**Table 1.** *Cont.*

| Categorical Variables | Valid Patients | Type of Healthcare | | p |
|---|---|---|---|---|
| | | SUS (%) | PHCS (%) | |
| Family income Per Capita (Minimum Wage) | | | | |
| ≤1 | | 196 (76.9) | 52 (16.2) | |
| >1 and ≤3 | 576 | 54 (21.1) | 162 (50.5) | |
| >3 ≤5 | | 3 (1.2) | 47 (14.6) | <0.001 |
| >5 | | 2 (0.8) | 60 (18.7) | |
| ABEP Classification | | | | |
| Class A | 50 | 3 (1.17) | 47 (14.46) | |
| Classes B1 and B2 | 179 | 26 (10.16) | 153 (47.08) | |
| Class C1 and C2 | 207 | 101 (39.45) | 106 (32.62) | <0.001 |
| Classes D–E | 145 | 126 (49.22) | 19 (5.85) | |
| ACS Classification | | | | |
| UA | | 20 (7.8) | 81 (24.9) | |
| NSTEMI | 581 | 47 (18.4) | 166 (51.1) | <0.001 |
| STEMI | | 189 (73.8) | 78 (24.0) | |
| Systemic Arterial Hypertension | 581 | 194 (75.8) | 270 (83.1) | 0.037 |
| Diabetes Mellitus | 581 | 76 (29.7) | 132 (40.6) | 0.008 |
| Dyslipidemia | 581 | 104 (40.6) | 218 (67.1) | <0.001 |
| Overweight | 576 | 153 (60.5) | 237 (73.4) | 0.001 |
| Abdominal Obesity | 568 | 171 (68.1) | 257 (81.1) | <0.001 |
| Sedentary lifestyle | 581 | 131 (51.2) | 180 (55.4) | 0.353 |
| Alcoholism | 581 | 39 (15.2) | 31 (9.5) | 0.049 |
| Smoking | | | | |
| No | | 100 (39.1) | 168 (51.7) | |
| Yes | 581 | 63 (24.6) | 36 (11.1) | <0.001 |
| Ex-smoker | | 93 (36.3) | 121 (37.2) | |
| Cardiovascular outcomes at 180 days after ACS3 | | 45 (17.6) | 54 (16.6) | 0.845 |
| Acute Coronary Syndrome | | 32 (12.5) | 36 (11.1) | 0.689 |
| Stroke | 581 | 5 (2.0) | 4 (1.2) | 0.516 |
| Congestive heart failure | | 7 (2.7) | 8 (2.5) | 0.987 |
| Cardiac Arrest | | 1 (0.4) | 6 (1.8) | 0.141 |
| Adherence to Physical Activity at 180 days after ACS | | | | |
| Sedentary | 488 | 133 (63.0) | 147 (53.1) | |
| Active | | 78 (37.0) | 130 (46.9) | 0.034 |
| Adherence to pharmacotherapy at 180 days after ACS | | | | |
| No | 488 | 88 (41.7) | 73 (26.4) | |
| Yes | | 123 (58.3) | 204 (73.6) | 0.001 |
| Smoking Cessation | | | | |
| Yes | 488 | 14 (6.6) | 11 (4.0) | |
| No | | 197 (93.4) | 266 (96.0) | 0.264 |
| Diet Quality at 180 days after ACS A | 488 | 47.79 (7.90) | 53.71 (8.98) | <0.001 |
| Hospitalization Time (days)A | 581 | 11.44 (11.6) | 9.42 (10.6) | <0.001 |

ACS = Acute Coronary Syndrome; SUS = Public Health Care; PHCS = Private Health Care System; ABEP = Brazilian Association of Research Companies [26]; UA = Unstable Angina; NSTEMI = Acute Myocardial Infarction without ST-segment elevation; STEMI = Acute Myocardial Infarction with ST-segment elevation; $p$ = Fisher's exact test or Pearson's chi-square; 1-Classification by body mass index [30]; 2-Classification by waist circumference [30]; 3-Total number of patients admitted to the study since new outcomes could arise during the ACS hospitalization; A = Mann–Whitney test: mean ± standard deviation.

In general, patients' QoL worsened, regardless of the type of healthcare at 30 days after the acute event, except for the emotional aspect. At 180 days after ACS, patients showed improvement in pain, social, and emotional aspects, with worsening of their functional capacity and general health status, compared with those during hospitalization (Table 2).

**Table 2.** Quality of life, according to SF–36 domains in patients from public and private health care systems who presented with ACS, Aracaju, Brazil.

| SF–36 Domains | Hospitalization | | 30 Days after ACS | | 180 Days after ACS | | p [D] |
|---|---|---|---|---|---|---|---|
| | Mean | SD | Mean | SD | Mean | SD | |
| *Functional Capacity* | 54.1 [A] | 32.0 | 36.9 [C] | 23.4 | 49.5 [B] | 25.0 | <0.001 |
| **Physical** *Aspect* | 41.3 [A] | 42.5 | 4.5 [B] | 11.0 | 40.4 [A] | 36.6 | <0.001 |
| Pain | 47.7 [B] | 30.0 | 40.3 [C] | 19.2 | 63.0 [A] | 14.8 | <0.001 |
| *General Health Status* | 57.3 [A] | 22.3 | 53.0 [C] | 19.1 | 54.8 [B] | 17.5 | 0.002 |
| *Vitality* | 59.9 [A] | 24.3 | 53.9 [B] | 17.6 | 62.4 [A] | 13.3 | <0.001 |
| *Social Aspect* | 67.8 [B] | 29.0 | 57.4 [C] | 19.8 | 79.9 [A] | 17.0 | <0.001 |
| *Emotional Aspect* | 59.8 [C] | 44.2 | 64.0 [B] | 40.7 | 83.6 [A] | 30.1 | <0.001 |
| *Mental Health* | 68.2 [A] | 22.5 | 64.7 [B] | 17.3 | 69.9 [A] | 12.7 | <0.001 |

ACS = Acute Coronary Syndrome; SD = standard deviation; D = Friedman test: multiple comparison; Wilcoxon test: comparison in pairs (Hospitalization versus 30 days after ACS; Hospitalization versus 180 days after ACS; 30 days after ACS versus 180 days after ACS). Equal letters indicate similar means ($p \geq 0.05$) and different letters indicate different means ($p < 0.05$), with the first letters of the alphabet (A, B, C) accompanying the highest means.

Table 3 shows the QoL of patients by type of healthcare. At admission, patients from the SUS had a higher mental health score than those from the private health care system. However, we verified an inverse situation for the emotional aspect. At 30 days after the acute event, patients in the public health care group had lower QoL in terms of physical aspect and pain. Compared with the QoL of patients from the public health care group at 180 days after ACS, the QoL of the patients in the private health care group was superior to all aspects addressed.

**Table 3.** Means and standard deviation of SF-36 domains of patients with ACS, according to the type of healthcare, Aracaju, Brazil.

| SF–36 Domains | Time of Evaluation | Type of Healthcare | | p |
|---|---|---|---|---|
| | | SUS Mean (±SD) | PCHS Mean (±SD) | |
| *Functional Capacity* | Hospitalization | 54.5 (32.2) | 53.7 (32.7) | 0.781 |
| | 30 days after ACS | 35.2 (22.5) | 38.2 (24.0) | 0.130 |
| | 180 days after ACS | 46.8 (23.8) | 51.6 (25.7) | 0.021 |
| *Physical Aspect* | Hospitalization | 40.5 (41.7) | 41.9 (43.2) | 0.871 |
| | 30 days after ACS | 2.7 (9.4) | 6.0 (12.0) | <0.001 |
| | 180 days after ACS | 31.5 (32.9) | 47.2 (37.8) | <0.001 |
| *Pain* | Hospitalization | 45.8 (32.0) | 49.2 (28.3) | 0.074 |
| | 30 days after ACS | 36.4 (18.2) | 43.4 (19.4) | <0.001 |
| | 180 days after ACS | 58.4 (13.9) | 66.5 (14.4) | <0.001 |
| *General Health Status* | Hospitalization | 56.9 (23.1) | 57.6 (21.7) | 0.778 |
| | 30 days after ACS | 52.0 (18.9) | 53.7 (19.2) | 0.310 |
| | 180 days after ACS | 53.0 (17.1) | 56.2 (17.7) | 0.043 |
| *Vitality* | Hospitalization | 61.8 (24.6) | 58.5 (23.9) | 0.102 |
| | 30 days after ACS | 54.2 (6.4) | 54.0 (18.5) | 0.971 |
| | 180 days after ACS | 60.6 (12.4) | 63.8 (13.8) | <0.001 |
| *Social Aspect* | Hospitalization | 70.1 (28.9) | 65.9 (29.0) | 0.062 |
| | 30 days after ACS | 56.9 (18.9) | 57.9 (20.6) | 0.362 |
| | 180 days after ACS | 78.5 (16.2) | 81.0 (17.5) | 0.022 |
| *Emotional Aspect* | Hospitalization | 53.5 (45.3) | 64.7 (42.6) | 0.003 |
| | 30 days after ACS | 60.0 (41.7) | 67.1 (39.6) | 0.064 |
| | 180 days after ACS | 80.4 (32.0) | 86.0 (28.3) | 0.027 |
| *Mental Health* | Hospitalization | 70.4 (22.5) | 66.6 (22.4) | 0.023 |
| | 30 days after ACS | 64.8 (17.0) | 64.6 (17.5) | 0.919 |
| | 180 days after ACS | 68.5 (13.2) | 71.0 (12.1) | 0.033 |

ACS = Acute Coronary Syndrome; SUS = Public Health Care; PHCS = Private Health Care System; SD = standard deviation; p = Mann–Whitney test.

When investigating patients' perception of their current health compared with that of a year ago, no distinction was found between the groups at the time of hospitalization. However, patients from the public health care group had a worsened perception about their own health compared to patients from the private health care group system, at 30 and 180 days after ACS (Table 4).

**Table 4.** Perception of patients with ACS concerning their current health compared to a year ago, according to the type of healthcare, Aracaju, Brazil.

| Time of Evaluation | Variables | Type of Healthcare | | *p* |
|---|---|---|---|---|
| | | SUS (%) | PHCS (%) | |
| *Hospitalization* | Much better | 37 (14.5) | 36 (11.1) | 0.111 |
| | A little better | 50 (19.5) | 45 (13.8) | |
| | Almost the same | 67 (26.2) | 108 (33.2) | |
| | A little worse | 79 (30.9) | 112 (34.5) | |
| | Much worse | 23 (9.0) | 24 (7.4) | |
| *30 Days after ACS* | Much better | - | - | 0.014 |
| | A little better | 3 (1.03) | 17 (5.9) | |
| | Almost the same | 106 (46.1) | 147 (50.9) | |
| | A little worse | 113 (49.1) | 118 (40.8) | |
| | Much worse | 8 (3.5) | 7 (2.4) | |
| *180 Days after ACS* | Much better | - | - | 0.008 |
| | A little better | 19 (9.0) | 31 (11.2) | |
| | Almost the same | 92 (43.6) | 156 (56.3) | |
| | A little worse | 92 (43.6) | 85 (30.7) | |
| | Much worse | 8 (3.8) | 5 (1.8) | |

ACS = Acute Coronary Syndrome; SUS = Public Health Care; PHCS = Private Health Care System; *p* = Pearson's chi-square.

In the multiple linear regression models, the best QoL of patients at 180 days after ACS was mainly associated with male sex and adherence to physical activity for all domains. Moreover, better SF-36 scores were found among individuals with shorter hospital stays, younger age, higher educational level, those who received medical treatment by the private health care system, the ones who did not develop subsequent cardiovascular events, those who had no history of SAH or DLP, and displayed adherence to pharmacotherapy (Table 5).

**Table 5.** Multiple linear regression models for QOL of patients at 180 days after ACS, Aracaju, Brazil.

| FUNCTIONAL CAPACITY ($r^2$ = 0.50) | | | | |
|---|---|---|---|---|
| Variables | β | CI (95%) | Standard Error | *p* |
| *Hospitalization time in days (Log)* | −2.31 | −4.55; −0.07 | 1.14 | 0.043 |
| *Age (years)* | −0.57 | −0.73; −0.41 | 0.08 | <0.001 |
| *Male Sex* | 15.97 | 12.35; 19.59 | 1.84 | <0.001 |
| *Schooling (years)* | 0.22 | −0.20; 0.65 | 0.22 | 0.301 |
| *Private Health Care System* | 7.22 | 2.95; 11.50 | 2.17 | 0.001 |
| *Systemic Arterial Hypertension* | −7.01 | −11.41; −2.61 | 2.24 | 0.002 |
| *Diabetes Mellitus* | −2.71 | −6.37; 0.94 | 1.86 | 0.146 |
| *Dyslipidemia* | −0.04 | −3.60; 3.52 | 1.81 | 0.983 |
| *Overweight* | 0.97 | −3.79; 5.72 | 2.42 | 0.690 |
| *Abdominal Obesity* | −0.25 | −5.37; 4.86 | 2.60 | 0.922 |
| *Cardiovascular Event* | −9.30 | −14.65; −3.95 | 2.72 | 0.001 |
| *Adherence to Physical Activity* | 19.68 | 16.20; 23.17 | 1.78 | <0.001 |
| *Adherence to Diet* | 0.91 | −2.45; 4.27 | 1.71 | 0.595 |
| *Adherence to Medication* | 0.64 | −3.00; 4.29 | 1.85 | 0.729 |
| *Smoking* | 1.85 | −5.75; 9.45 | 3.87 | 0.633 |

**Table 5.** *Cont.*

| PHYSICAL ASPECT ($r^2$ = 0.34) | | | | |
|---|---|---|---|---|
| **Variables** | **β** | **CI (95%)** | **Standard Error** | ***p*** |
| *Hospitalization time in days (Log)* | −5.71 | −9.41; −2.01 | 1.88 | 0.003 |
| *Age (years)* | −0.21 | −0.47; 0.06 | 0.14 | 0.124 |
| *Male Sex* | 14.36 | 8.37; 20.34 | 3.05 | <0.001 |
| *Schooling (years)* | 1.25 | 0.55; 1.96 | 0.36 | 0.001 |
| *Private Health Care System* | 10.10 | 3.04; 17.16 | 3.59 | 0.005 |
| *Systemic Arterial Hypertension* | −8.67 | −15.94; −1.39 | 3.70 | 0.020 |
| *Diabetes Mellitus* | −3.24 | −9.28; 2.81 | 3.08 | 0.293 |
| *Dyslipidemia* | −0.38 | −6.26; 5.51 | 3.00 | 0.900 |
| *Overweight* | −1.19 | −9.05; 6.66 | 4.00 | 0.765 |
| *Abdominal Obesity* | −2.41 | −10.85; 6.04 | 4.30 | 0.576 |
| *Cardiovascular Event* | −11.98 | −20.82; −3.13 | 4.50 | 0.008 |
| *Adherence to Physical Activity* | 26.64 | 20.88; 32.40 | 2.93 | <0.001 |
| *Adherence to Diet* | 3.66 | −1.89; 9.21 | 2.82 | 0.195 |
| *Adherence to Medication* | −0.08 | −6.10; 5.95 | 3.06 | 0.980 |
| *Smoking* | 15.32 | 2.76; 27.88 | 6.39 | 0.087 |
| PAIN ($r^2$ = 0.15) | | | | |
| **Variables** | **β** | **CI (95%)** | **Standard Error** | ***p*** |
| *Hospitalization time in days (Log)* | −1.65 | −3.34; 0.03 | 0.86 | 0.055 |
| *Age (years)* | −0.16 | −0.28; −0.04 | 0.06 | 0.011 |
| *Male Sex* | 2.82 | 0.09; 5.54 | 1.39 | 0.043 |
| *Schooling (years)* | −0.06 | −0.38; 0.26 | 0.16 | 0.716 |
| *Private Health Care System* | 8.54 | 5.33; 11.76 | 1.64 | <0.001 |
| *Systemic Arterial Hypertension* | −2.36 | −5.67; 0.96 | 1.69 | 0.163 |
| *Diabetes Mellitus* | 2.54 | −0.22; 5.29 | 1.40 | 0.071 |
| *Dyslipidemia* | −0.54 | −3.22; 2.14 | 1.36 | 0.691 |
| *Overweight* | 2.26 | −1.32; 5.84 | 1.82 | 0.216 |
| *Abdominal Obesity* | 2.03 | −1.81; 5.88 | 1.96 | 0.300 |
| *Cardiovascular Event* | −1.43 | −5.46; 2.60 | 2.05 | 0.486 |
| *Adherence to Physical Activity* | 5.83 | 3.21; 8.46 | 1.34 | <0.001 |
| *Adherence to Diet* | 0.54 | −1.99; 3.07 | 1.29 | 0.674 |
| *Adherence to Medication* | −0.22 | −2.96; 2.53 | 1.40 | 0.877 |
| *Smoking* | 2.09 | −3.64; 7.81 | 2.91 | 0.474 |
| GENERAL HEALTH STATUS ($r^2$ = 0.19) | | | | |
| **Variables** | **β** | **CI (95%)** | **Standard Error** | ***p*** |
| *Hospitalization time in days (Log)* | −4.31 | −6.28; −2.34 | 1.00 | <0.001 |
| *Age (years)* | −0.01 | −0.15; 0.13 | 0.07 | 0.875 |
| *Male Sex* | 5.30 | 2.11; 8.48 | 1.62 | 0.001 |
| *Schooling (years)* | 0.48 | 0.11; 0.86 | 0.19 | 0.011 |
| *Private Health Care System* | 1.24 | −2.52; 4.99 | 1.91 | 0.517 |
| *Systemic Arterial Hypertension* | −5.79 | −9.66; −1.92 | 1.97 | 0.003 |
| *Diabetes Mellitus* | −0.15 | −3.36; 3.07 | 1.64 | 0.928 |
| *Dyslipidemia* | −3.58 | −6.71; −0.45 | 1.59 | 0.025 |
| *Overweight* | 3.68 | −0.50; 7.86 | 2.13 | 0.084 |
| *Abdominal Obesity* | 3.96 | −0.53; 8.46 | 2.29 | 0.084 |
| *Cardiovascular Event* | −4.37 | −9.08; 0.34 | 2.40 | 0.069 |
| *Adherence to Physical Activity* | 6.04 | 2.97; 9.11 | 1.56 | <0.001 |
| *Adherence to Diet* | 0.22 | −2.73; 3.17 | 1.50 | 0.883 |
| *Adherence to Medication* | 0.34 | −2.87; 3.54 | 1.63 | 0.837 |
| *Smoking* | −3.96 | −10.64; 2.73 | 3.40 | 0.245 |

**Table 5.** *Cont.*

| VITALITY (r² = 0.14) | | | | |
|---|---|---|---|---|
| **Variables** | **β** | **CI (95%)** | **Standard Error** | ***p*** |
| *Hospitalization time in days (Log)* | −1.30 | −2.84; 0.25 | 0.79 | 0.101 |
| *Age (years)* | 0.00 | −0.11; 0.11 | 0.06 | 0.964 |
| *Male Sex* | 4.31 | 1.81; 6.82 | 1.27 | 0.001 |
| *Schooling (years)* | 0.15 | −0.15; 0.44 | 0.15 | 0.320 |
| *Private Health Care System* | 2.03 | −0.92; 4.98 | 1.50 | 0.178 |
| *Systemic Arterial Hypertension* | −1.47 | −4.51; 1.57 | 1.55 | 0.343 |
| *Diabetes Mellitus* | 0.63 | −1.90; 3.15 | 1.29 | 0.627 |
| *Dyslipidemia* | −1.21 | −3.67; 1.25 | 1.25 | 0.333 |
| *Overweight* | 2.13 | −1.15; 5.42 | 1.67 | 0.202 |
| *Abdominal Obesity* | −0.67 | −4.21; 2.86 | 1.80 | 0.708 |
| *Cardiovascular Event* | −5.10 | −8.79; −1.40 | 1.88 | 0.007 |
| *Adherence to Physical Activity* | 5.53 | 3.12; 7.94 | 1.23 | <0.001 |
| *Adherence to Diet* | −0.55 | −2.87; 1.77 | 1.18 | 0.642 |
| *Adherence to Medication* | 2.72 | 0.20; 5.24 | 1.28 | 0.034 |
| *Smoking* | −0.29 | −5.54; 4.96 | 2.67 | 0.913 |
| SOCIAL ASPECT (r² = 0.13) | | | | |
| **Variables** | **β** | **CI (95%)** | **Standard Error** | ***p*** |
| *Hospitalization time in days (Log)* | −1.97 | −3.94; 0.01 | 1.01 | 0.051 |
| *Age (years)* | −0.12 | −0.26; 0.02 | 0.07 | 0.092 |
| *Male Sex* | 5.73 | 2.52; 8.93 | 1.63 | <0.001 |
| *Schooling (years)* | 0.39 | 0.76; 0.01 | 0.19 | 0.044 |
| *Private Health Care System* | 4.57 | 0.79; 8.34 | 1.92 | 0.018 |
| *Systemic Arterial Hypertension* | −2.79 | −6.68; 1.10 | 1.98 | 0.159 |
| *Diabetes Mellitus* | −0.14 | −3.37; 3.09 | 1.65 | 0.932 |
| *Dyslipidemia* | −1.00 | −4.15; 2.15 | 1.60 | 0.534 |
| *Overweight* | −0.70 | −4.90; 3.50 | 2.14 | 0.744 |
| *Abdominal Obesity* | −3.69 | −8.21; 0.82 | 2.30 | 0.109 |
| *Cardiovascular Event* | −8.14 | −12.87; −3.41 | 2.41 | 0.001 |
| *Adherence to Physical Activity* | 6.49 | 3.41; 9.57 | 1.57 | <.001 |
| *Adherence to Diet* | 0.70 | −2.27; 3.66 | 1.51 | 0.644 |
| *Adherence to Medication* | 0.58 | −2.65; 3.80 | 1.64 | 0.726 |
| *Smoking* | −1.13 | −7.84; 5.59 | 3.42 | 0.742 |
| EMOTIONAL ASPECT (r² = 0.15) | | | | |
| **Variables** | **β** | **CI (95%)** | **Standard Error** | ***p*** |
| *Hospitalization time in days (Log)* | −2.98 | −6.42; 0.45 | 1.75 | 0.089 |
| *Age (years)* | −0.10 | −0.34; 0.15 | 0.13 | 0.447 |
| *Male Sex* | 8.83 | 3.27; 14.39 | 2.83 | 0.002 |
| *Schooling (years)* | −0.18 | −0.84; 0.47 | 0.33 | 0.580 |
| *Private Health Care System* | 5.73 | −0.83; 12.28 | 3.34 | 0.087 |
| *Systemic Arterial Hypertension* | −2.37 | −9.13; 4.39 | 3.44 | 0.491 |
| *Diabetes Mellitus* | −3.20 | −8.82; 2.42 | 2.86 | 0.263 |
| *Dyslipidemia* | 0.02 | −5.45; 5.48 | 2.78 | 0.995 |
| *Overweight* | 3.67 | −3.63; 10.97 | 3.71 | 0.324 |
| *Abdominal Obesity* | 1.46 | −6.39; 9.31 | 3.99 | 0.715 |
| *Cardiovascular Event* | −16.84 | −25.06; −8.62 | 4.18 | <0.001 |
| *Adherence to Physical Activity* | 13.76 | 8.40; 19.11 | 2.72 | <0.001 |
| *Adherence to Diet* | 2.57 | −2.59; 7.72 | 2.62 | 0.329 |
| *Adherence to Medication* | 4.78 | −0.81; 10.37 | 2.85 | 0.094 |
| *Smoking* | 3.83 | −7.84; 15.49 | 5.94 | 0.520 |

**Table 5.** *Cont.*

| MENTAL HEALTH ($r^2 = 0.10$) | | | | |
|---|---|---|---|---|
| **Variables** | **β** | **CI (95%)** | **Standard Error** | ***p*** |
| *Hospitalization time in days (Log)* | −0.29 | −1.81; 1.23 | 0.77 | 0.706 |
| *Age (years)* | 0.07 | −0.04; 0.17 | 0.06 | 0.246 |
| *Male Sex* | 4.10 | 1.64; 6.55 | 1.25 | 0.001 |
| *Schooling (years)* | 0.00 | −0.29; 0.29 | 0.15 | 0.983 |
| *Private Health Care System* | 2.63 | −0.27; 5.52 | 1.48 | 0.075 |
| *Systemic Arterial Hypertension* | −1.24 | −4.22; 1.75 | 1.52 | 0.416 |
| *Diabetes Mellitus* | −0.22 | −2.70; 2.27 | 1.26 | 0.865 |
| *Dyslipidemia* | −1.98 | −4.40; 0.43 | 1.23 | 0.107 |
| *Overweight* | −0.22 | −3.45; 3.00 | 1.64 | 0.891 |
| *Abdominal Obesity* | −1.23 | −4.7; 2.24 | 1.76 | 0.487 |
| *Cardiovascular Event* | −5.87 | −9.50; −2.24 | 1.85 | 0.002 |
| *Adherence to Physical Activity* | 3.56 | 1.19; 5.92 | 1.20 | 0.003 |
| *Adherence to Diet* | −0.87 | −3.15; 1.40 | 1.16 | 0.452 |
| *Adherence to Medication* | 2.02 | −0.46; 4.49 | 1.26 | 0.110 |
| *Smoking* | −3.57 | −8.72; 1.59 | 2.62 | 0.175 |

QoL = Quality of Life; ACS = Acute Coronary Syndrome; CI = Confidence Interval; Log = Logarithm; Multiple linear regression models were performed considering: Period of hospitalization; Age at interview; Gender: 0 = Female and 1 = Male; Schooling in the interview; Type of Healthcare: 0 = SUS and 1 = Private health care system; Systemic Arterial Hypertension: 0 = No and 1 = Yes; Diabetes Mellitus: 0 = No and 1 = Yes; Dyslipidemia: 0 = No and 1 = Yes; Overweight: 0 = No and 1 = Yes; Abdominal obesity: 0 = No and 1 = Yes; Cardiovascular event at 180 days after ACS: 0 = No and 1 = Yes; Adherence to physical activity: 0 = Sedentary and 1 = Active; Better diet quality: 0 = No and 1 = Yes; Adherence to medication: 0 = No and 1 = Yes; Smoking: 0 = No (Adherence) and 1 = Yes (No adherence).

## 4. Discussion

In this study, in general, the patients' QoL improved in only three of the eight SF-36 domains 180 days after ACS. Individuals assisted by the private health care network showed better QOL for all domains of the SF-36 when compared to those assisted by the public service. We also found that the better QoL was associated with the male sex and adherence to physical activity for all the evaluated components.

At 180 days after ACS, we associated the absence of a subsequent cardiovascular event and access to the private health care system with higher scores for the six and four SF-36 domains (functional capacity, physical appearance, pain, and general health, and mental, which covers vitality, social aspects, emotional state, and mental health), respectively. We also associated a shorter hospital stay, lower age group, higher educational level, absence of SAH and DLP, and adherence to pharmacotherapy with better QoL.

Studies reported that improvement in QoL is an outcome of aftercare practices, serving as a basis for decision-making in public health policies [11]. Therefore, the results of this study are relevant when considering that the QoL of patients with worse results, after 180 days of ACS, may be associated with longer hospitalization time, shorter adherence to secondary prevention guidelines performed at hospital discharge, and to the public health model. These data were independent predictors of these findings.

We verified that 30 days after ACS, there was a reduction in the scores of seven SF-36 domains, mainly in the ones related to the physical component. However, this may result from the post-hospitalization due to ACS. Thus, we will center our discussion on the results found 180 days after the acute event.

AMI is a highly stressful life-threatening disease that may have consequences on patient well-being for a substantial time, with limited physical functioning, cardiac complications, and deterioration of QoL [31]. Literature shows worse QoL in those who experienced cardiovascular events compared to their healthy counterparts [32].

At 180 days after ACS, QoL improved in only one of the physical components (pain) and worsened in two of them (functional capacity and general health). In a study conducted with ACS patients to verify changes in their QoL and their functional capacity, researchers detected that 8 months after hospital discharge, their functional capacity declined [33].

These data are similar to our study results, where we verified that functional capacity at the previous levels before the acute event was not recovered.

When assessing patients' QoL by type of healthcare, during hospitalization, similarities were found between the groups in six SF-36 domains. At 180 days after ACS, when compared with patients in the private health care group, those from the public group (SUS) had worse QoL for all assessed items and had worsened perception about their own health. A longer hospitalization time and lower rates of adherence to secondary prevention treatment (less adherence to physical activity, medication therapy, and lower diet quality) in patients from the public health care group (SUS) suggest possible distortions in health care quality between the two groups. Considering these differences in the assistance received and the socioeconomic characteristics intrinsic to patients from the SUS, these factors had a negative influence on the results, culminating in worse QoL scores for this group. These data are consistent with the literature in showing that health care quality and socioeconomic context can influence treatment adherence [16,17], prognosis, and patients' QoL [13,15].

The results showed an association between worse QoL and increased age, female sex, lower educational level, and a higher prevalence of comorbidities. Three of these characteristics (older age, female sex, and higher prevalence of comorbidities) were more frequent in patients from the private health care group. Despite this, patients who received medical treatment by this service had better QoL, leading once again to questions about the health care models adopted in Sergipe and Brazil as a whole, especially when considering that studies show the negative impact of chronic conditions on worsening QoL in individuals, which is more accentuated with multiple comorbidities [34,35].

The low adherence to secondary prevention by the study patients is a possible sign of health care distortions in Brazil. The benefits of secondary prevention therapies in patients with ACS are evident [36,37], as is the fact that individuals with better adherence to this therapy in intervention studies showed a reduction in hospital readmission rates, cardiovascular mortality, improved health [1] and QOL [38,39].

Therefore, these results have implications for public health policies in Sergipe and, possibly, in Brazil, showing that strategies for improving health care quality are fundamental to create mechanisms for better adherence to secondary prevention and, consequently, better QoL in patients after ACS.

Our analysis had some limitations. First, the advanced vascular unit (UVA in Portuguese) from the public hospital included in this study interrupted patient care in July 2014 and June 2015, contributing to a smaller number of patients treated at this service. Second, results were limited by information on adherence to pharmacotherapy as well as smoking cessation or persistence because we collected these data with simple self-report questions, without using validated measuring instruments.

However, we believe that this study is one of the first to be conducted in Brazil to compare QoL data after hospital discharge in patients with ACS and compare different types of health care. This shows that social factors and possible disparities in health care quality influence QoL after ACS in Sergipe. However, we can speculate that the results presented here reflect the general situation in Brazil.

## 5. Conclusions

In conclusion, patients receiving medical treatment by the private health care system had better QoL than patients receiving medical care by the public health care system (from SUS), showing a disparity in health care quality. This is a challenge that we must overcome to improve the efficiency and equitability of the health care system.

**Author Contributions:** I.M.N.B.d.C.C., D.G.d.S. and A.C.S.S. conceptualized the study, coordinated the last follow-up with participants from the study, conducted the statistical analysis, and wrote the manuscript. I.M.N.B.d.C.C., D.G.d.S. and A.C.S.S. conceptualized the study and wrote the manuscript. J.R.S.S. conducted the statistical analysis. J.L.M.O., F.A.d.A., J.d.G.J., L.M.S.M.d.O., R.R.d.A., J.O.C., M.F.C.d.S., L.M.C.P., L.V.S.A., S.M.V., M.A.A.-S., F.J.A., V.B.O., L.S.M. and L.B. wrote the manuscript. Supervision: A.C.S.S. All authors have read and agreed to the published version of the manuscript.

**Funding:** This research was funded by the Higher Education Personnel Improvement Coordination (CAPES, Brazil) n° 1793619.

**Institutional Review Board Statement:** The study was conducted following the Declaration of Helsinki, and approved by Research Ethics Committee involving human beings at the Federal University of Sergipe (CEP/UFS). The committee approved our research (approval no. 302,544/date: 06-07-2013). for studies involving humans.

**Conflicts of Interest:** The authors declare no conflict of interest.

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
