# Peer review of "Quality of Life among Patients with Acute Coronary Syndromes Receiving Care from Public and Private Health Care Systems in Brazil"

_clinpract, doi:10.3390/clinpract12040055_

Round 1

Reviewer 1 Report

The manuscript by Ingrid Maria Novais Barros de Carvalho Costa et al. entitled “Life Quality of Patients with Acute Coronary Syndrome Receiving Public and Private Health Care System” aimed to analyze the QOL of patients with acute coronary syndrome (ACS) who were assisted by a public or private health care system.

The abstract summarizes the general significance of the manuscript. Moreover, the article leads some evidence to such point, but the exclusion criteria are unclear. and there are many cofounding variables; consequently,  some major issues need to be addressed to improve the significance of the manuscript:

-Firstly, the different classes of drugs taken by patients should be reported. Were the patients all taking optimal medical therapy?

-Also, the main echocardiographic parameters should be reported, such as TAPSE and LVEF. Please, add these parameters in Tables.

-Another crucial aspect is the program of rehabilitation exercises: did patients in the two groups undergo the same rehabilitation program?

- Finally, it would be essential to perform sub-analyzes based on the type of ACS.

Author Response

I hope this one finds you well. Thank you for your considerations and all statements were accepted and the text was adequate.

Reviewer 2 Report

The authors of this article entitled ‘‘Life quality of patients with acute coronary syndrome receiving public and private health care system’’ aimed to explore the quality of life among a Brazilian population of patients with acute coronary syndromes who received medical care from public and private health care systems. The authors try to provide insight into the factors affecting the quality of life of patients after an ACS and assess whether the discrepancies in the management of patients with ACS between public and private health care systems in Brazil have an impact on the patients’ quality of life. Below, several remarks are presented to the authors that need to be addressed.

1. Lines 2-3: Please revise the title. There are syntax-expression errors that render the current title futile. You should also specify in the title that the article involves patients from Brazil. Suggestion: ‘‘Quality of life among patients with acute coronary syndromes receiving care from public and private health care systems in Brazil’’

2. The manuscript is not reader friendly, since in many parts there is lack of coherence in terms of writing flow. One may need to read certain sentences several times in order to comprehend the meaning.

3. There are multiple expression and editing issues.

4. Several parts of the manuscript should undergo thorough English editing.

5. The inclusion and exclusion criteria of the study should be better defined.

6. Lines 98-121: There are sentences missing (probably hidden) in the boxes of Figure 1. Please correct.

7. Since this was a prospective study, why wasn’t a power calculation/sample size estimation performed prior to study commencement?

8. How were the variables included in the multiple linear regression models selected? The authors need to mention their criteria and the number of variables their sample size would allow them to include.

9. How do you explain that the best QOL was associated with male sex?

10. Please provide an explanation as to why the emotional aspect showed constant improvement after hospitalization for ACS.

11. It is not specified whether the same medical and interventional protocols were used in all 4 hospitals, i.e. if all patients with STEMI underwent primary PTCA, if they received the same medical treatment during hospitalization, if the same medication was administered for secondary prevention, etc.

12. Was the follow-up of the patients the same in terms of implementation of the secondary prevention measures? During the 180 days after discharge, were both arms of the patients (from public and private health care system) handled in the same way (follow-up in outpatient clinics at regular intervals, same opportunities with doctor counseling etc.)?

13. Data may not reflect current status in Sergipe since authors started collecting them almost a decade ago. (October 2013 to March 2016)

14. Several of the references should be revised, since some are outdated, while others are rather irrelevant in terms of supporting the manuscript’s statements.

15. Lines 371-372: Please use the most recent 2017 ESC guidelines for STEMI with regard to Reference 9.

16. Lines 350-351: Please revise Reference 1.

17. Line 62: Reference 11 is irrelevant, since it does not correspond to this part of the sentence, and should be placed after the phrase ‘‘improvement in the quality of life (QOL) of these patients’’

18. Line 63: Reference 14 does not support the aforementioned sentence. The same holds true for reference 13.

19. Line 77: References 13 and 15 are considered irrelevant to the following sentence ‘‘Moreover, evidence shows that disunity in the quality of health care may have a negative influence on treatment adherence [22], compromising the prognosis and QOL of patients [13,15].’’

20. Line 207: Please provide a more relevant reference instead of reference 35.

21. Lines 288-291: ‘‘In a study conducted with ACS patients to verify changes in their QOL and their functional capacity, researchers detected that at 8 months after the hospitalization discharge, their functional capacity declined [38].’’ Please revise, since the design, aim and conclusions of the study cited as reference 38 are considered irrelevant to what you are trying to point out.

22. Line 305: References 13 and 15 are considered irrelevant.

23. In Table 1, data about family income should be properly aligned. The same should be done about adherence to physical activity and pharmacotherapy at 180 days after ACS

24. In Table 1, when talking about smoking cessation, I believe that you should use only data from smokers and the variables should be ‘‘yes’’ or ‘‘no’’ and not ‘‘smokers’’ and ‘‘non-smokers’’. Please explain.

25. In Table 1, you should explain to the readers what ABEP classification is for, which I understand is a socioeconomic classification.

26. In the legend of Table 1, please explain the meaning of ‘‘3-Total number of patients admitted to the study since new outcomes could arise during the ACS hospitalization’’

27. Line 264: It would be nice to specify again, in the discussion part, the 3 SF-36 domains in which QOL showed improvement, so that the information is readily available to the reader who will not need to look back at the results section . The same should be done in Line 270.

28. Several commas throughout the text are redundant. I would suggest not using Oxford comma, since it can sometimes lead to confusion and negatively affect the normal flow of reading. The use of Oxford comma can be found throughout the text e.g. Line 73 (equity,) / Line 157 (pain,) / Line 158 (state,) / Line 182 (overweight,) / Line 200 (outcomes,) / Line 201 (cessation,) / Line 212 (social,) / Line 285 (complications,) /Line 305 (prognosis,) and so on

29. Line 38: Please revise the phrase ‘‘were assisted’’. Possible suggestions: received medical treatment, received medical care, were treated, were handled

30. Line 39: Please omit the word ‘‘and’’

31. Line 40: I would suggest replacing the phrase ‘‘in cardiology’’ with ‘‘providing cardiology services’’.

32. Line 42: I would rephrase the phrase ‘‘according to the type of medical assistance’’ to ‘‘according to the type of health care provided’’.

33. Line 43: Please correct the word ‘‘model’’ to ‘‘models’’.

34. Line 44: ‘‘A total of 581 patients were eligible, including 44.1% and 55.9% in the public and private groups, respectively.’’ Please rephrase.

35. Line 45: Prior to reporting that ‘‘the public health care group had lower QOL scores for all domains’’, there should have been a previous short reference that SF-36 addresses certain domains.

36. Line 48: ‘‘This shows that social factors and possible disparities in health care quality influence QOL after ACS in Sergipe.’’ Please revise so that the word quality is not mentioned twice.

37. Lines 53-57: ‘‘Despite the evidence in disparities, advances in the diagnosis and treatment of patients with ACS have led to a significant increase in the number of survivors of an acute event, constituting a challenge for health care systems to accomplish the need to offer effective and equitable secondary prevention measures.’’ Please revise.

38. Line 58: ‘‘Brazilian2,8 and international9,10 guidelines…’’ Please put references in brackets.

39. Line 58: I would suggest using the phrase ‘‘emphasize the importance’’ instead of ‘‘point the importance’’

40. Lines 60-62: ‘‘… and treatment adherence to control risk factors (RF) and comorbidities, contributing to increased survival [8,9,10,11] and improvement in the quality of life (QOL) of these patients.’’ Please revise.

41. Line 64: Please correct ‘‘and considered’’ to ‘‘and is considered’’

42. Line 72: Please add the article ‘‘ a’’ in the phrase ‘‘is predominantly private system’’, i.e. ‘‘is predominantly a private system’’

43. Line 73: When using ‘‘universalization’’, I assume you mean universality.

44. Line 73: I am not sure about the use of the word ‘‘integrality’’

45. Line 74: ‘‘However, there are reports of disparity between private and public health care on the appropriate treatment of patients with ACS’’. I would suggest slightly changing the sentence as follows: ‘‘However, there are reports of existing disparities between private and public health care with regard to the appropriate treatment of patients with ACS’’

46. Line 75: I am not sure about the use of the word ‘‘disunity’’ in the phrase ‘‘disunity in the quality of health care’’. Maybe the use of ‘‘discrepancy’’ or ‘‘heterogeneity’’ would be more appropriate. i.e. ‘‘discrepancies in the quality of health care provided’’

47. Lines 77-79: ‘‘However, there is no representative knowledge about the QOL of patients with ACS that are SUS users or the presence of disparity compared with those assisted by the private health care system.’’ Please revise.

48. Line 82: Please omit the word ‘‘and’’

49. Line 83: I would suggest replacing the phrase ‘‘in cardiology’’ with ‘‘providing cardiology services’’.

50. Lines 84-85: ‘‘does not have an “open-door” service, which requires referral of patients from another health institution.’’. I suppose you intend to say ‘‘does not have an “open-door” service, which means that it requires referral of patients from another health institution.’’

51. Line 86: The acronym PHCS should correspond to the phrase ‘‘Private Health Care Services’’ instead of ‘‘Private Services’’.

52. Line 100: Please correct ‘‘17 death’’ to ‘‘17 died’’ or ‘‘17 deaths’’

53. Line 101: What do you mean that ‘‘4 had ACS after hospitalization’’? Do you mean recurrent ACS? If so, why were they excluded from the study, since in your analysis you actually include ACS in the studied variables as one of the cardiovascular events that occurred within 180 days after the initial ACS?

54. Line 127: Please correct ‘‘or AMI with ST-segment elevation (NSTEMI)’’ to ‘‘or AMI with ST-segment elevation (STEMI)’’

55. Lines 138-139: ‘‘Data were collected from October 2013 to March 2016, with respect to the individual characteristics of the institutions.’’. Could you please specify what you mean by the phrase ‘‘with respect to the individual characteristics of the institutions.’’?

56. Line 140: I believe that stage 2 and 3 should be presented separately in different parentheses, since you are analyzing the 3 stages

57. Line 142: I would recommend replacing ‘‘show information’’ with ‘‘provide information’’.

58. Line 144: Please replace ‘‘the patient’’ with ‘‘the patients’’ in the sentence ‘‘when the patient could not respond to the questionnaire by themselves.’’

59. Line 146: ‘‘On admission’’ instead of ‘‘at admission’’

60. Line 147: ‘‘the International Physical Activity Questionnaire (short version)25,26 was used to assess…’’ Please put references in brackets.

61. Line 149: I would put a comma before the word ‘‘adapted’’ i.e. ‘‘the Alternative Healthy Eating Index (2010) [28], adapted from the Food Guide for the Brazilian Population [21,29], was used…’’. Moreover, the particular sentence needs to be rephrased, since I assume you mean to say that the Alternative Healthy Eating Index was adapted to the Food Guide for the Brazilian Population.

62. Line 151: Please correct ‘‘the higher the values, the better state of health’’ to ‘‘the higher the values, the better the state of health’’

63. Lines 151-153: ‘‘At 180 days after ACS, patients were asked about the use of medications prescribed at hospital discharge, smoking cessation, and information on new cardiovascular events.’’ Please rephrase.

64. Line 155: ‘‘ (SF-36) questionnaire [17,30], since we used it in research with a specific focus on cardiology [12,31].’’ Please revise.

65. Line 160: Please replace ‘‘shows’’ with another word (e.g. ‘‘includes’’)

66. Line 160: Please correct the phrase ‘‘the individual perception of’’ to ‘‘the individual’s perception of’’

67. Lines 161-162: ‘‘It consists of a self-administered instrument, which can also be part of an interview, whether face-to-face or by telephone’’. It is rather vague whether this sentence refers to SF-36 as a whole or to the assessment of the individual’s perception of their own health. If it refers to SF-36 as a whole, then you should change the order of the sentences for purposes of coherence in writing flow.

68. Lines 168-169: ‘‘according to the health care model received when they presented ACS.’’ Please revise.

69. Lines 171: Please correct ‘‘frequency’’ to ‘‘frequencies’’.

70. Line 180: Please replace ‘‘type of health assistance’’ with ‘‘type of health care’’

71. Line 187: ‘‘with 5% of statistical significance’’. Please rephrase.

72. Line 190: Please revise the phrase ‘‘were assisted’’.

73. In Table 1 please correct ‘‘familiar income’’ to ‘‘family income’’.

74. In Table 1, I assume you mean sedentary lifestyle by ‘‘sedentarism’’.

75. Line 216: Please correct the phrase ‘‘in patients from public and private health care system who presented ACS’’ to ‘‘in patients from public and private health care systems who presented with ACS’’

76. Line 218: Please correct ‘‘SCA’’ to ‘‘ACS’’.

77. Line 223: ‘‘On admission’’ instead of ‘‘at admission’’

78. In Tables 3 and 4 please replace ‘‘moment of the evaluation’’  with ‘‘time of evaluation’’ 

79. Line 237: I would suggest using the word ‘‘difference’’ instead of ‘‘distinction’’

80. Line 250: Please correct ‘‘lower age’’ to ‘‘younger age’’. The same should be done in Line 271.

81. Lines 250-251: ‘‘better SF-36 scores were found among individuals with shorter hospital stays, lower age, higher educational level, assisted by private health care system, without subsequent cardiovascular events, SAH, DLP, and adherence to pharmacotherapy’’. For purposes of clarity the above sentence should be revised as follows:  : ‘‘better SF-36 scores were found among individuals with shorter hospital stays, younger age, higher educational level, who received medical treatment by the private health care system, developed no subsequent cardiovascular events, had no history of SAH or DLP and displayed adherence to pharmacotherapy’’. Another option would be to divide the sentence into 2 smaller sentences.

82. Lines 265 and 266: Please revise the word ‘‘assisted’’.

83. Lines 274-278: ‘‘Therefore, the results of this study are more relevant when considering the evidence of a worse QOL of patients 180 days after ACS, longer hospitalization time, less adherence to secondary prevention guidelines performed at hospital discharge, and the public health care model. These data were independent predictors of these findings.’’ Please revise.

84. Line 286: I would replace the phrase ‘‘cardiovascular diseases’’ with ‘‘cardiovascular events’’

85. Line 290: Please correct the phrase ‘‘after the hospitalization discharge’’ to ‘‘after hospital discharge’’

86. Line 292: I would change the phrase ‘‘at the previous levels of the acute event’’ to ‘‘at the previous levels before the acute event’’.

87. Line 296: I would use the phrase ‘‘worse perception of their own health’’ instead of ‘‘worsened perception about their own health’’

88. Line 299: I would recommend using the word ‘‘discrepancies’’ instead of ‘‘distortions’’

89. Lines 301 and 309: Please revise the word ‘‘assistance’’ and ‘‘assisted’’.

90. Line 315-318: ‘‘The benefits of secondary prevention therapies in patients with ACS are unequivocal [42,43], as well the fact that individuals with better adherence to this therapy in intervention studies showed a reduction in hospital readmission rates, cardiovascular mortality, higher levels of health,7 and QOL [44].’’ Please revise.

91. Line 318: ‘‘higher levels of health,7 …’’ Please put the reference in brackets.

92. Line 318: Please specify the meaning of ‘‘higher levels of health’’

93. Lines 329-331: ‘‘However, we believed that this study is one of the first studies carried out in Brazil to compare QOL data after hospital discharge in patients with ACS and received public versus private assistance.’’ Please revise.

94. Lines 335 and 336: Please revise the word ‘‘assisted’’.

95. Line 338: I am not sure about the use of the word ‘‘equitability’’ 

Author Response

(The authors gave the same response as above.)

Round 2

Reviewer 1 Report

I thank the authors for the answers; however, the article still has many gaps.

Author Response

Dear reviewer,

We would like, initially, to thank you for the careful reading of our manuscript. We consider all the issues raised to be relevant. See also the text to verify the incorporation of suggestions, in which the changes are highlighted.

Please see the attachment,

Rebeca Rocha

Reviewer 2 Report

I appreciate the fact that the authors tried to provide information about some ambiguous parts in the manuscript and made some corrections. However, there are still issues that need to be addressed. Proper editing is lacking in several parts, there are still many expression and syntax issues, some references are not appropriate. More attention from the authors’ standpoint would be desirable before having proceeded to re-submission of this revised version. In detail:

1. ‘‘Despite the progress in the diagnosis and treatment of patients with ACS, which have contributed to a significant increase in the number of survivors after an acute event, it is still a challenge for health systems to provide effective, equitable secondary prevention measures [3,4,5,6] and addressing disparities in health care system for these patients.’’ There are grammatical, syntax and expression issues in the sentence.

2. ‘’Prognosis and clinical evolution of patients after hospital discharge can be modified based on the therapy adopted and compliance to treatment, contributing to a reduction and control of risk factors (RF) and comorbidities, collaborating to an increase in survival [7, 8, 9] and improvement in the quality of life (QoL) of these patients [10]. ‘’ The sentence needs to be revised, since there are expression issues.

3. ‘’However, information is scarce in the literature on the QOL of patients with ACS assisted in the SUS or private health care, and on the presence of disparity between health care systems is scarce in the literature.’’ Again the sentence needs to be revised, since there are syntax errors and expression issues.

4. The current reference 7 should be referring to the 2017 ESC guidelines for STEMI and not to the 2021 ESC guidelines for heart failure. This has already been suggested to the authors in the previous review report (Comment 15, Lines 371-372: Please use the most recent 2017 ESC guidelines for STEMI with regard to Reference 9.)

5. The current reference 9 (previous reference 11) is still in the wrong place and has not been put in the correct part of the sentence, although it has already been suggested to the authors in the previous review report (Comment 17, Line 62: Reference 11 is irrelevant, since it does not correspond to this part of the sentence, and should be placed after the phrase ‘‘improvement in the quality of life (QOL) of these patients’’. Answer: We thank you for the consideration, realized in the text.)

6. ‘’Moreover, evidence shows that distortions in the quality of health care may have a negative influence on treatment adherence, compromising the prognosis and QoL of patients [18].’’ Instead of distortions, I believe that the word ‘’discrepancies’’ or ‘’heterogeneity’’ is more appropriate.

7.  In the revised manuscript, in Figure 1, there are still sentences missing (probably hidden) in the boxes. Please correct.

8. Regarding questions 9 (How do you explain that the best QOL was associated with male sex?) and 10 (Please provide an explanation as to why the emotional aspect showed constant improvement after hospitalization for ACS) from the previous review report, the answers provided by the authors should be incorporated, along with the corresponding references, in the Discussion part.

9. ‘’In a study conducted with ACS patients to verify changes in their QoL and their functional capacity, researchers detected that 8 months after hospital discharge, their functional capacity declined [31].’’ Reference 31 does not support the above mentioned sentence. In fact, it is totally irrelevant. Please place the correct reference.

10. ‘’These data are consistent with the literature in showing that health care quality and socioeconomic context can influence treatment adherence [16,17], prognosis, and patients’ QoL [13,15].’’ Instead of reference 13, please provide a more relevant reference.

11. Regarding former comment 26 (In the legend of Table 1, please explain the meaning of ‘‘3-Total number of patients admitted to the study since new outcomes could arise during the ACS hospitalization’’), please rephrase the certain sentence, so that it is clearly comprehensible to all readers, like you did in your answer to my comment. The purpose of my comments is not just to explain what you mean to me (the reviewer), but to improve your overall manuscript so that the readers can easily understand what you are saying.

12. Comment 27 from the previous review report (Line 264: It would be nice to specify again, in the discussion part, the 3 SF-36 domains in which QOL showed improvement, so that the information is readily available to the reader who will not need to look back at the results section) has not been addressed by the authors . Regarding Line 270, please specify in separate parentheses the 6 and 4 SF-36 domains, since in the parenthesis that has been added 8 domains are mentioned.

13. Comment 52 (Line 100: Please correct ‘‘17 death’’ to ‘‘17 died’’ or ‘‘17 deaths’’) has not been addressed by the authors.

14. Regarding former comment 53 (Line 101: What do you mean that ‘‘4 had ACS after hospitalization’’? Do you mean recurrent ACS? If so, why were they excluded from the study, since in your analysis you actually include ACS in the studied variables as one of the cardiovascular events that occurred within 180 days after the initial ACS?), the authors should revise the sentence ‘‘4 had ACS after hospitalization’’ according to the answer they have provided.

15. ‘’At admission and 180 days after ACS, the International Physical Activity Questionnaire (short version)[20, 21] was used to assess adherence to physical activity recommendations.’’ Please use ‘‘On admission’’ instead of ‘‘at admission’’ [it has already been proposed in former comment 59 (Line 146)]

16. Regarding the sentence ‘’the Alternative Healthy Eating Index (2010) [23] adapted from the Food Guide for the Brazilian Population [23,24] was used to assess their diet quality…’’, I insist that the phrase ‘’adapted from’’ is not proper, since it conveys the exact opposite meaning from what you want to say. By using this phrase, it is as if the Food Guide for the Brazilian Population served as the base to devise the Alternative Healthy Eating Index, which is not the case.

17. Comment 64 [Line 155: ‘‘ (SF-36) questionnaire [17,30], since we used it in research with a specific focus on cardiology [12,31].’’ Please revise.] has not been addressed by the authors.

18. ‘’SF-36 also includes an item that assesses the individuals' perception …’’. Please correct ‘’ the individuals’ ’’  to ‘’the individual’s’’ [It has already been  suggested in former comment 66-Line 160: Please correct the phrase ‘‘the individual perception of’’ to ‘‘the individual’s perception of’’]

19. Former comment 68 (Lines 168-169: ‘‘according to the health care model received when they presented ACS.’’ Please revise.) has not been addressed by the authors. The phrase ‘’they presented ACS’’ should be revised to ‘’ they presented with ACS’’ or ‘’ they developed ACS’’, depending on what you mean to say.

20. ‘’At admission, patients from the SUS had a higher mental health score than those from the private health care system.’’ Please use ‘‘On admission’’ instead of ‘‘at admission’’ [it has already been proposed in former comment 77 (Line 223)]

21. Regarding the sentence ‘’When investigating patients’ perception of their current health compared with that a year ago, no distinction was found between the groups at the time of hospitalization.’’ ,  former comment 79 (Line 237: I would suggest using the word ‘‘difference’’ instead of ‘‘distinction’’. Answer: We thank you for the consideration, realized in the text.) has not been addressed by the authors.

22. Regarding the sentence ‘’Individuals assisted by the private health care network showed better QOL for all domains of the SF-36 when compared to those assisted by the public service.’’ , former comment 82 (Lines 265 and 266: Please revise the word ‘‘assisted’’. Answer: We thank you for the consideration, realized in the text.) has not been addressed by the authors.

23. Regarding the sentence ‘‘At 180 days after ACS, when compared with patients in the private health care group, those from the public group (SUS) had worse QoL for all assessed items and had worsened perception about their own health.’’, former comment 87 (Line 296: I would use the phrase ‘‘worse perception of their own health’’ instead of ‘‘worsened perception about their own health’’. Answer: We thank you for the consideration, realized in the text.) has not been addressed by the authors.

24. Regarding the sentence ‘’… in patients from the public health care group (SUS) suggest possible distortions in health care quality between the two groups.’’ , former comment 88 (Line 299: I would recommend using the word ‘‘discrepancies’’ instead of ‘‘distortions’’. Answer: We thank you for the consideration, realized in the text.) has not been addressed by the authors.

25. Regarding former comment 89 (Lines 301 and 309: Please revise the word ‘‘assistance’’ and ‘‘assisted’’. Answer: Sorry, we could not identify that word in the indicated lines), since the authors couldn’t find the indicated lines, here is the sentence in Line 301 I am referring to:

‘’Considering these differences in the assistance received and the socioeconomic characteristics intrinsic to patients from the SUS, these factors had a negative influence on the results, culminating in worse QoL scores for this group.’’

Line 309 has been properly addressed by the authors.

26. The following sentence ‘’However, we believe that this study is one of the first to be conducted in Brazil to compare QoL data after hospital discharge in patients with ACS and compare different types of health care.’’ needs to be properly revised in terms of expression and syntax.

27. ‘’In the context of secondary prevention, some classes of medications are labeled as A according to the Specialized Guidelines…’’. I suppose you mean that some medications have a class IA recommendation. Please revise.

28. Please correct Prazygrel to Prasugrel.

29. Please revise the sentence ‘’We collected the data related to the prescriptions of the medicines mentioned above from the medical records and compared them to the prescriptions, with the patients present at the moment of hospital discharge.’’

30. ‘’About the socioeconomic level of the sample, according to the Brazilian Economic Classification Criterion of the Brazilian Association of Research Companies (ABEP) [XX].’’ Please provide a complete sentence. Also, the reference needs to be properly written.

31. Please correct the sentence ‘’ For purposes of analysis, the eight economic levels, or levels, or economic classes, established by ABEP, were were regrouped and named as follows…’’.

32. ‘’SF-36 consists of a self-administered instrument, which can also be part of an interview, whether face-to-face or by telephone [28], and SF-36 is composed of 36 questions that address eight domains in two major components: physical, which involves functional capacity, physical appearance, pain, and general health, and mental, which covers vitality, social aspects, emotional state, and mental health.’’ Please revise. It would be better to break them into two smaller sentences.

33. In the sentence ‘’Therefore, a face-to-face interview was carried out at admission and by telephone at 30 and 180 days after the acute event’’. please use ‘‘on admission’’ instead of ‘‘at admission’’

34. In the sentence ‘’No distinction was found between the groups regarding the occurrence of cardiovascular outcomes at 180 days of ACS’’, I would suggest using the word ‘‘difference’’ instead of ‘‘distinction’’.

35. ‘’Patients treated by the private health care system had more comorbidities, but with a shorter hospital stay’’. Please revise the sentence in terms of expression.

36. Please use appropriate spaces in Table 1 for the following: ‘’from 50 to59’’ and ‘’from 70 to79’’

37. There should be appropriate blank lines that separate the legends of the tables with the text that follows. i.e.

- after ‘’A = Mann-Whitney test: mean ± standard deviation.’’ there should be a blank line that separates it from the following text ‘’ In general, patients’ QoL worsened, regardless of the type of healthcare at 30 days after the acute event, except for the emotional aspect.’’

- after ‘’… capacity and general health status, compared with those during hospitalization (Table 2).’’ there should be a blank line separating it from the following  ‘’Table 2 Quality of life, according to SF – 36 domains in patients from public and private health care systems…’’

-after the footnotes in Table 2 ‘’… accompanying the highest means.’’, there should be a blank line that separates it from the following text ‘’ Table 3 shows the QoL of patients by type of healthcare.’’

-after ‘’… the private health care group was superior to all aspects addressed.’’, there should be a blank line separating it from the following ‘’Table 3 Means and standard deviation of SF – 36…’’

-after the footnotes in Table 3 ‘’ p=Mann-Whitney test.’’ , there should be a blank line that separates it from the following text ‘’When investigating patients’ perception of their…’’

38. ‘’We also found that the better QoL was associated with the male sex and adherence to physical activity for all the evaluated components.’’ Please revise.

39. ‘’lower age group’’. Please revise

40. ‘’shorter adherence to secondary prevention guidelines’’. Please revise

41. ‘’Therefore, the results of this study are relevant when considering that the QoL of patients with worse results, after 180 days of ACS, may be associated with longer hospitalization time, shorter adherence to secondary prevention guidelines performed at hospital discharge, and to the public health model.’’ Please revise

42. Although I disagree with the use of Oxford commas, because it can sometimes lead to confusion and negatively affect the normal flow of reading, the authors have chosen to maintain Oxford commas throughout the text. However, this is only just a minor comment of trivial importance.

Author Response

Dear Reviewer,

 We would like, initially, to thank you for the careful reading of our manuscript. We consider all the issues raised to be relevant. See also the text to verify the incorporation of suggestions, in which the changes are highlighted.
